

# Using present-day observations to detect when anthropogenic change forces surface ocean carbonate chemistry outside pre-industrial bounds

Adrienne J. Sutton[1,2], Christopher L. Sabine[2], Richard A. Feely[2], Wei-Jun Cai[3], Meghan F. Cronin[2], Michael J. McPhaden[2], Julio M. Morell[4], Jan A. Newton[5], Jae-Hoon Noh[6], Sólveig R. Ólafsdóttir[7], Joseph E. Salisbury[8], Uwe Send[9], Douglas C. Vandemark[8], Robert A. Weller[10]

[1]Joint Institute for the Study of the Atmosphere and Ocean, University of Washington, Seattle, WA, 98195, USA
[2]Pacific Marine Environmental Laboratory, NOAA, Seattle, WA, 98115, USA
[3]School of Marine Science and Policy, University of Delaware, Newark, DE, 19716, USA
[4]Department of Marine Sciences, University of Puerto Rico, Mayagüez, 00681, Puerto Rico
[5]Applied Physics Laboratory, University of Washington, Seattle, WA, 98105, USA
[6]Korea Institute of Ocean Science and Technology, Ansan Gyunggido, 15627, Korea
[7]Marine Research Institute, Skulagata 4, 101 Reykjavik, Iceland
[8]Ocean Processes Analysis Laboratory, University of New Hampshire, Durham, NH, 03825, USA
[9]Scripps Institution of Oceanography, University of California, San Diego, La Jolla, CA, 92093, USA
[10]Woods Hole Oceanographic Institution, Woods Hole, MA, 02543, USA

Correspondence to: A. J. Sutton (adrienne.sutton@noaa.gov)

**Abstract.** One of the major challenges to assessing the impact of ocean acidification on marine life is the need to better understand the magnitude of long-term change in the context of natural variability. This study addresses this need through a global synthesis of monthly pH and aragonite saturation state ($\Omega_{arag}$) climatologies for 12 open ocean, coastal, and coral reef locations using 3-hourly moored observations of surface seawater partial pressure of $CO_2$ and pH collected together since as early as 2010. Mooring observations suggest open ocean subtropical and subarctic sites experience present-day surface pH and $\Omega_{arag}$ conditions outside the bounds of pre-industrial variability throughout most, if not all, of the year. In general, coastal mooring sites experience more natural variability and thus, more overlap with pre-industrial conditions; however, present day $\Omega_{arag}$ conditions surpass biologically relevant thresholds associated with ocean acidification impacts on *Mytilus californianus* ($\Omega_{arag}$<1.8) and *Crassostrea gigas* ($\Omega_{arag}$<2.0) larvae in the California Current Ecosystem (CCE) and *Mya arenaria* larvae in the Gulf of Maine ($\Omega_{arag}$<1.6). At the most variable mooring locations in coastal systems of the CCE, subseasonal conditions approached $\Omega_{arag}$=1. Global and regional models and data syntheses of ship-based observations tended to underestimate seasonal variability compared to mooring observations. Efforts such as this to characterize all modes of pH and $\Omega_{arag}$ variability and change at key locations are fundamental to assessing present-day biological impacts of ocean acidification, further improving experimental design to interrogate organism response under real-world conditions, and improving predictive models and vulnerability assessments seeking to quantify the broader impacts of ocean acidification.

## 1 Introduction

The global ocean has absorbed ~30% of anthropogenic carbon dioxide ($CO_2$) emissions released since the beginning of the industrial revolution (Le Quéré et al., 2015; Sabine et al., 2004). Although ocean uptake of $CO_2$ has played a role in mitigating the atmospheric-associated impacts of anthropogenic $CO_2$, it has also resulted in changes to seawater chemistry. Seawater pH has decreased globally by 0.1 since the pre-industrial era (Feely et al., 2004; Orr et al., 2005) and is predicted to decrease by another 0.3 by 2100 under Intergovernmental Panel on Climate Change (IPCC) business-as-usual emission scenarios (Caldeira and Wickett, 2005; Orr et al., 2005). Ocean acidification also results in reductions in carbonate ion concentrations and the saturation




states of calcium carbonate minerals utilized by calcifying marine organisms to make their shells or skeletons. Globally, average surface seawater pH change is currently about -0.002 yr$^{-1}$ and the saturation state of aragonite, a common form of calcium carbonate mineral utilized by marine organisms, is approximately -0.008 yr$^{-1}$ (Bates et al., 2014; Takahashi et al., 2014).

These assessments of ocean-wide change inform global-scale predictions of ocean acidification impacts and provide boundaries on carbonate chemistry in designing biological experiments. However, a major challenge to assessing local- to regional-level ecological and economic consequences of ocean acidification is the lack of understanding of how global anthropogenic change manifests relative to natural variability, especially in dynamic coastal regions (Bauer et al., 2013). Coastal systems are sites of large variability, where terrestrial, atmospheric, and marine nutrient and carbon cycles all interact and where ocean acidification
conditions can be more extreme than open ocean environments (Cai et al., 2011; Feely et al, 2010). Important processes that affect ocean acidification in the coastal ocean include upwelling, riverine/estuarine input, air-sea gas exchange, production and respiration, calcification, dissolution, sediment burial, and sea-ice dynamics. Despite the ecological, biological, and economic importance of coastal regions, the magnitude and variability of these key biogeochemical processes are poorly quantified.

Earth system models provide some insights into carbon system variability; however, they do not often capture the full magnitude of variability, especially at the seasonal to subseasonal time scales (Pilcher et al., 2015; Sasse et al., 2015). For example, Sasse et al. (2015) estimated that earth system models underpredict the seasonal cycle of seawater partial pressure of $CO_2$, i.e., $p(CO_2)$, by 30%. Despite these biases, some studies have made progress utilizing current estimates of $\Omega_{arag}$ variability and change in assessing ecosystem impact and vulnerability to ocean acidification. Using predicted aragonite saturation state ($\Omega_{arag}$) change between 2005
and 2050 and an estimate of annual variability from the Community Climate System Model (CCSM) 3.1, Cooley et al. (2009) identified that ocean acidification will exceed natural conditions throughout the global ocean by 2050, especially in low-latitude regions of the Atlantic, Indian, and western Pacific. Applying similar approaches to three other earth system models, Friedrich et al. (2012) also concluded these regions are most vulnerable, but also found that present-day ocean acidification already exceeds pre-industrial variability by a factor of 5 in shallow water tropical Pacific and Atlantic coral reef ecosystems. Both studies pointed
out that these global models do not currently resolve coastal processes and therefore lack important sources of natural variability, which bias these results when extrapolating to coastal and coral systems. In a vulnerability assessment of U.S. shellfisheries, this lack of coastal information was addressed by using earth system model output as a baseline for $\Omega_{arag}$ conditions but also adding a term in the final assessment for amplification of ocean acidification in coastal systems that experience eutrophication, upwelling, and river inputs of low $\Omega_{arag}$ water (Ekstrom et al., 2015). Although this approach did not resolve all coastal carbonate variability
and change, it allowed for some of the first regional-level assessments of ocean acidification impact. Regional models in coastal systems are also showing promise at eddy-resolving spatial scales and monthly to seasonal temporal scales. Applied to the California Current Ecosystem (CCE), regional models predict surface ocean $\Omega_{arag}$ has already moved outside the bounds of pre-industrial variability (Hauri et al., 2013).

Direct measurements of the full range of variability will help parameterize and evaluate these models. They can also help improve our understanding of the baseline, pre-industrial conditions to which marine life adapted. Understanding this range of pre-industrial variability can be used to investigate whether organisms are currently stressed outside of their natural range, and if so, what the biological impacts are of that stress. Understanding what dominant modes of variability define this natural range is also critical to future predictions of impact. Whether the dominant modes are stochastic events (e.g., storms), the seasonal cycle, interannual
variability (e.g., El Niño and La Niña events), or decadal climactic oscillations (e.g., Pacific Decadal Oscillation) will control how





climate change impacts these modes of variability and associated ocean carbonate chemistry. Knowledge about when and where corrosive conditions occur and how the timing of such conditions may relate to key life stages is also critical to assessing vulnerability to ocean acidification. By resolving these scales of variability, researchers can start to refine biological experiments, ecosystem models, and economic vulnerability assessments in the context of full ecosystem variability and change. As evidence

emerges that some shelled marine organisms are already being impacted by corrosive seawater chemistry conditions (Barton et al., 2012; Bednaršek et al., 2014a; Bednaršek et al., 2014b; Bednaršek et al., 2012; Reum et al., 2015; Salisbury et al., 2008) and stakeholders are seeking locally relevant solutions (Kelly and Caldwell, 2013; Kelly et al., 2011), these analyses are becoming increasingly important.

High-frequency moored observations can be highly effective in capturing the full range of variability at key locations (Cullison Gray et al., 2011; Harris et al., 2013; Hofmann et al., 2011; Reimer et al., in review; Shadwick et al., 2015; Sutton et al., 2014a; Xue et al., 2016). Here we present the first global-scale ocean acidification mooring synthesis of 3-hourly surface ocean pH observations and $\Omega_{arag}$ calculated from direct measurements of $p(CO_2)$ and pH collected together on 12 open ocean, coastal, and coral reef moorings since as early as 2010. We used these observations to define present-day and pre-industrial subseasonal to

interannual pH and $\Omega_{arag}$ variability at each location and to ground truth carbonate chemistry variability in earth system models. This assessment identifies the patterns in which present-day conditions have exceeded pre-industrial bounds and biologically relevant thresholds at these mooring locations, and documents when and where marine life currently encounter ocean acidification conditions that may impact growth and survival.

## 2 Methods

In 2003, the National Oceanographic and Atmospheric Administration's (NOAA) Pacific Marine Environmental Laboratory (PMEL) began to establish a global moored $CO_2$ network that presently includes 38 mooring locations (www.pmel.noaa.gov/co2/). With the recent development of autonomous pH sensors, PMEL has been able to expand this network to include pH at 21 of the mooring locations. These moored $CO_2$ and ocean acidification time series are part of a long-term, sustained effort intended to advance our scientific understanding of the ocean carbon cycle and how it is changing over time. This network leverages other

observing system efforts including, but not limited to, the National Data Buoy Center, the Research Moored Array for African-Asian-Australian Monsoon Analysis and Prediction, and the OceanSITES Network. This study focuses on 12 of these moorings with paired $p(CO_2)$ and pH measurements that allow for estimates of subseasonal to interannual variability of pH and $\Omega_{arag}$ (Fig. 1, Table 1). This set of moorings includes sites in each major ocean basin and in a variety of open ocean, coastal, and coral reef environments. The time series used in this study include observations primarily between 2010 and 2015, with the exception of the

WHOTS time series, which goes back to 2004. "Present day" observations used here refer to these time series outlined in Table 1.

Each mooring time series summarized in Table 1 included a Moored Autonomous $p(CO_2)$ (MAPCO₂) system and a Sunburst SAMI-pH sensor deployed on the surface buoy. For a detailed description of the MAPCO₂ system and data processing, refer to Sutton et al. (2014b). In brief, the MAPCO₂ system utilizes an automated equilibrator-based gas collection system to measure

surface seawater $x(CO_2)$ (the mole fraction of $CO_2$ in air in equilibrium with surface seawater) every 3 hours in addition to sample temperature, pressure, and relative humidity. The $x(CO_2)$ measurement is made by a nondispersive infrared gas analyzer (LI-820, LI-COR) calibrated before, during, and after field deployment with reference gases traceable to World Meteorological Organization standards. A Sea-Bird Electronics (SBE) 16plus V2 SeaCAT was also deployed and integrated with the MAPCO₂ system to collect sea surface temperature (SST) and salinity (SSS) measurements used to calculate $p(CO_2)$ consistent with ocean





$CO_2$ standard operating procedures (Dickson et al., 2007; Weiss, 1974). Overall uncertainty of the $MAPCO_2$ is <2 µatm for seawater $p(CO_2)$. The SAMI-pH sensor utilizes the spectrophotometric method for measuring seawater pH with a laboratory-based accuracy of ±0.003 and precision of <0.001 (Seidel et al., 2008).

The SAMI-pH and SBE SeaCAT sensors collected 3-hourly measurements during the $MAPCO_2$ 10 min seawater equilibration time. All sensors were factory calibrated between each 1 year deployment. Data quality control for the SAMI-pH data involved utilizing the sensor software to identify and flag outliers and sensor failures such as bad blanks. We also used the relationship between total alkalinity ($A_T$) and SSS (Cullison Gray et al., 2011; Fassbender, 2014; Lee et al., 2006; Xue et al., 2016) to calculate pH from $p(CO_2)$ and $A_T$ (using the CO2SYS program) in order to identify sensor drift within a deployment or offsets between

deployments, which only occurred in 8% of the pH data sets. All pH data are on the total pH scale (Dickson et al., 2007). Data quality control for the SBE SeaCAT was limited to flagging and eliminating outliers. All measurements are archived at the relevant data centers: the Carbon Dioxide Information Analysis Center (http://cdiac.ornl.gov/oceans/Moorings/) and the National Centers for Environmental Information (https://www.ncei.noaa.gov/).

In all calculations of $\Omega_{arag}$ we used the MATLAB version (v1.1) of the CO2SYS program (Lewis and Wallace, 1998; van Heuven et al., 2011) with the carbonic acid dissociation constants of Lueker et al. (2000), sulfate dissociation constants of Dickson (1990), and borate-to-salinity ratio of Lee et al. (2010) according to recommended best practices (Dickson et al., 2007; Orr et al., 2015). We also used average surface ocean phosphate and silicate concentrations from the World Ocean Atlas 2009 for each mooring location (Garcia et al., 2010). We used two pairs of carbonate system parameters to calculate $\Omega_{arag}$: 1) $p(CO_2)$ and pH observations

and 2) $p(CO_2)$ observations and $A_T$ estimated from SSS (Cullison Gray et al., 2011 for CCE2 and Chá bă; Fassbender, 2014 for Papa and KEO; C. Hunt, University of New Hampshire, personal communication, 2016 for Gulf of Maine; Xue et al., 2016 for Gray's Reef; and Lee et al., 2006 for the remaining sites). We then averaged the two $\Omega_{arag}$ data sets for the final present day $\Omega_{arag}$ values presented here. The only exception to the $\Omega_{arag}$ calculations was the WHOTS mooring time series where we have good pH data over multiple years but only during the months of June to November. For this site we used only $p(CO_2)$ and $A_T$ estimated

according to Lee et al. (2006) to calculate $\Omega_{arag}$. Similarly, all pH data presented are direct autonomous measurements of pH except in the case of the WHOTS mooring where pH was calculated from $p(CO_2)$ observations and estimated $A_T$ using CO2SYS.

    We used this approach of averaging calculated $\Omega_{arag}$ from $p(CO_2)$ and pH observations ($\Omega_{arag: pCO2-pH}$) and calculated $\Omega_{arag}$ from $p(CO_2)$ observations and estimated $A_T$ ($\Omega_{arag: pCO2-AT}$) to minimize the following errors: 1) the covariance of $p(CO_2)$ and pH can

lead to small errors in predicting $\Omega_{arag}$, which can enhance $\Omega_{arag}$ variability (Cullison Gray et al., 2011; Millero, 2007) and 2) the $A_T$ proxies only account for dilution and evaporation processes, which can result in underestimation of $A_T$ variability and, when combined with other carbon parameters, can reduce surface $\Omega_{arag}$ variability. We tested the $A_T$ proxies by comparing $A_T$ measured from discrete bottle samples collected at the surface at each mooring site to $A_T$ estimated from the proxies listed above, which were developed using $A_T$ and SSS measurements throughout the mixed layer. $A_T$ estimated from 3-hourly moored SSS was

generally within stated errors (3 to 20 µmol kg$^{-1}$) of discrete $A_T$ (e.g., Table 2). However, at a few sites where discrete $A_T$ was measured at the highest frequency (weekly to seasonally; sample size [n] > 100), discrete $A_T$ was more variable than $A_T$ estimated from the $A_T$-SSS proxies (Table 2). At the La Parguera (coral reef) and WHOTS (open ocean) mooring sites, variability of discrete $A_T$ (as measured by 1 standard deviation [SD] of the mean) was greater than estimated $A_T$ by 21% and 52%, respectively.





In addition to the discrepancies in discrete and proxy $A_T$ data sets, the SD, or variability, of both $\Omega_{arag}$ calculated from discrete measurements ($\Omega_{arag:\ discrete}$) and $\Omega_{arag:\ pCO2\text{-}pH}$ were greater than $\Omega_{arag}$ calculated from the $p(CO_2)$-$A_T$ and pH-$A_T$ pairs ($\Omega_{arag:\ pCO2\text{-}AT}$ and $\Omega_{arag:\ pH\text{-}AT}$; Table 2). One limitation of the WHOTS data set is that high-frequency, moored pH observations only exist during the months of June to November; thus, the mean and SD of the $\Omega_{arag:\ pCO2\text{-}pH}$ and $\Omega_{arag:\ pH\text{-}AT}$ data sets may be biased. Therefore, we

used La Parguera as a guide and assumed actual variability (i.e., from high-quality discrete bottle samples) of $\Omega_{arag}$ ($\Omega_{arag:\ discrete}$ SD = 0.16) lies between $\Omega_{arag:\ pCO2\text{-}pH}$ (SD = 0.19) and $\Omega_{arag}$ calculated from estimated $A_T$ ($\Omega_{arag:\ pCO2\text{-}AT}$ SD = 0.12; $\Omega_{arag:\ pH\text{-}AT}$ SD = 0.13; Table 2) and used the approach of averaging mooring ocean acidification data sets to reflect this. Since we are confident in the uncertainty of the moored autonomous $p(CO_2)$ measurements (Sutton et al., 2014b), for $\Omega_{arag}$ calculated from estimated $A_T$ we selected $\Omega_{arag:\ pCO2\text{-}AT}$ to average with $\Omega_{arag:\ pCO2\text{-}pH}$. While there were not enough discrete measurements to do this same comparison

at the coastal sites, we found that the $\Omega_{arag:\ pCO2\text{-}pH}$, $\Omega_{arag:\ pCO2\text{-}AT}$, and $\Omega_{arag:\ pH\text{-}AT}$ data sets were not significantly different ($p<0.05$) at the coastal sites, likely due to high natural variability in the coastal carbon system. This suggests that averaging $\Omega_{arag:\ pCO2\text{-}pH}$ and $\Omega_{arag:\ pCO2\text{-}AT}$ versus using one $\Omega_{arag}$ data set will not have a significant impact on results.

The ideal method for calculating $\Omega_{arag}$ may differ across sites dependent on access to and analysis of high-quality ship-based

measurements. However, averaging $\Omega_{arag:\ pCO2\text{-}pH}$ and $\Omega_{arag:\ pCO2\text{-}AT}$ provides a conservative estimate of monthly $\Omega_{arag}$ conditions allowing for a broad-scale comparison with consistent methodology across 12 buoys in different marine ecosystems. Continued high frequency (ideally, weekly to monthly where feasible) discrete sampling, development and analysis of regional $A_T$ proxies, and development of autonomous sensors capable of measuring carbonate parameters other than $p(CO_2)$ and pH will all help to further refine these methods for calculating $\Omega_{arag}$ from moored autonomous observations.

To estimate pre-industrial pH and $\Omega_{arag}$, we used $p(CO_2)$ and $A_T$ with the following adjustments to present-day observed $p(CO_2)$ and SST: 1) a decrease of 105 ppm in surface ocean $p(CO_2)$ assuming the delta between atmospheric and surface seawater $p(CO_2)$ has remained constant from the pre-industrial era to a reference year of 2010 (from: Dr. Pieter Tans, NOAA/ESRL, www.esrl.noaa.gov/gmd/ccgg/trends/), and 2) changes in SST that vary regionally from 0.5 to 1.5°C consistent with the IPCC Fifth

Assessment Report (IPCC, 2013). Although magnitudes of atmospheric $CO_2$ uptake can also vary regionally (Bauer et al, 2013), for the purposes of this synthesis we assumed that the change in air-sea $CO_2$ differences was consistent globally but that SST changes varied regionally as presented by the IPCC (2013). We also assumed small changes in SSS, $A_T$, phosphate, or silicate since the pre-industrial era would not have a significant impact on calculated pH and $\Omega_{arag}$. We applied these changes to monthly mean and range of variability of present-day observations and used CO2SYS to calculate pre-industrial pH and $\Omega_{arag}$. The range of

monthly variability in pre-industrial and present-day conditions were defined by descriptive statistics used in box and whisker plots excluding outliers:

$$Lower\ limit = Q1 - 1.5 \times IQR \qquad\qquad (1)$$
$$Upper\ limit = Q3 - 1.5 \times IQR \qquad\qquad (2)$$

where Q1 is the 25th percentile (or first quartile), Q3 is the 75th percentile (or third quartile), and IQR is the interquartile range (Q3-Q1) of pH and $\Omega_{arag}$ values. These limits equate to approximately ±2.7 SD of the mean for a normally distributed data set.

The variables used to estimate total uncertainty of present day calculated $\Omega_{arag}$ are shown in Table 3. This uncertainty estimate

included a preliminary assessment of in situ validation similar to $p(CO_2)$ validation by Sutton et al. (2014b) to estimate pH error



in the field. The average difference between SAMI-pH measurements and pH calculated from discrete measurements of dissolved inorganic carbon and $A_T$ was ±0.018, which was larger than laboratory-based assessments of pH measurement error (Seidel et al., 2008). While this estimate included error caused by slight mismatches in space (<1 km) and time (<1.5 hours) between the moored and discrete measurements, we used it here to develop a conservative estimate of total estimated uncertainty for calculated $\Omega_{arag:}$

$_{pCO2-pH}$ from moored observations, which is 0.37 for seawater at SST = 25°C, SSS = 35, $p$(CO₂) = 370 µatm, and pH = 8.1 (Table 3). This estimate meets the target relative uncertainty for $\Omega_{arag}$ of 10% needed to identify relative spatial patterns and short-term variation in ocean acidification (Newton et al., 2015). A more detailed assessment of pH sensor error is planned as more discrete and autonomous pH data become available. When the MAPCO₂ system is paired with estimated $A_T$, $\Omega_{arag: pCO2-AT}$ also meets this uncertainty target (Table 3); however, natural variability of surface ocean $\Omega_{arag: pCO2-AT}$ may be underestimated as discussed

previously. This estimated uncertainty is likely higher during the months of May through July at the Gulf of Maine mooring site where SSS <30 25% of the time due to freshwater inputs; at low salinity, the $A_T$–SSS relationship likely deteriorates. Total relative uncertainty of $\Omega_{arag}$ for all mooring data sets, averaged from the $\Omega_{arag: pCO2-pH}$ and $\Omega_{arag: pCO2-AT}$ data sets, likely falls between 5 and 10% (at the seawater conditions described in Table 3) and within the target relative uncertainty for describing short-term ocean acidification variability. From this point on, $\Omega_{arag}$ reported here is $\Omega_{arag: pCO2-AT}$ for the WHOTS mooring time series and averaged

$\Omega_{arag: pCO2-pH}$ and $\Omega_{arag: pCO2-AT}$ for the other mooring time series.

## 3 Results and Discussion

### 3.1 Observations of variability and change

Direct observations of $p$(CO₂) and pH revealed present-day conditions of surface ocean carbonate chemistry in 12 different oceanic and coastal systems. The open ocean mooring time series sites are located in subtropical oligotrophic regions (WHOTS, Stratus),

biologically productive subtropical regions that experience seasonal monsoons (BOBOA) and tropical cyclones (KEO), and subarctic regions with pronounced seasonality of physical and biological conditions (Papa, Iceland). Annual mean $\Omega_{arag}$ at these sites ranged from 1.83 to 3.56; annual mean pH ranged from 7.99 to 8.12 (Figs. 1–4, Table 4). High biological productivity is a feature at each of the four coastal mooring time series sites on the continental shelves of the U.S. East (Gulf of Maine, Gray's Reef) and West (CCE2, Chá bă) coasts. Summer upwelling is another important driver of conditions at Chá bă, located mid-shelf

at 100 m bottom depth offshore of La Push, Washington (Alford et al., 2012). While upwelling can also impact the CCE2 site located mid-shelf at 800 m bottom depth farther south in the CCE, this subregion in particular has shown sensitivity to climatic drivers, such as the El Niño Southern Oscillation (ENSO; Nam et al., 2011). Seasonal temperature and freshwater inputs impact natural variability at the two coastal moorings in the Atlantic with the Gulf of Maine site located 10 km from shore at 65 m bottom depth (Salisbury et al., 2009) and Gray's Reef 70 km from shore at 20 m bottom depth (Reimer et al., in review; Xue et al., 2016).

The Chuuk and La Parguera sites are located in coral reef ecosystems within a semi-closed atoll lagoon in Micronesia in 23 m bottom depth and a patch reef in the Caribbean Sea southwest of Puerto Rico in 5 m bottom depth, respectively. Despite the more variable coastal and coral reef conditions, the range of annual mean $\Omega_{arag}$ at these sites was 1.97 to 3.37, less than the range observed at the more diverse set of ocean regimes represented by the open ocean sites (Figs. 1, 5–7, Table 4). However, the range of annual mean pH was approximately the same as the open ocean sites from 8.01 to 8.15 (Figs. 5–7).

Of the open ocean time series, the moorings located in subtropical oligotrophic regions, WHOTS and Stratus, experienced lower seasonal to subseasonal variability in surface pH and $\Omega_{arag}$ (Figs. 2–4). Consistent trade winds and shallow mixed layer depth throughout the year along with the lack of deep winter convection likely contribute to this relatively low open ocean variability. Temporal variability was higher at the other four open ocean mooring locations, which was likely driven by 1) more prevalent



seasonal changes in SST (on average 2 times more variable than WHOTS and Stratus) and productivity, and 2) stochastic events such as storms and typhoons. In general, the range of variability tended to be consistent throughout the annual cycle at each of the open ocean sites with exceptions of increased variability at the Iceland location in late summer and early fall and at Papa during winter (Fig. 4). Present-day $\Omega_{arag}$ values were mostly >3 year-round at the subtropical open ocean sites except at Stratus, where

5  $\Omega_{arag}$ values mostly fell between 2.5 and 3.0 (Figs. 2 and 3). Surface $\Omega_{arag}$ conditions were further reduced at Papa and Iceland, the subarctic sites, which range from 1.5 to 2.5 (Fig. 4). Present-day pH observations were >8 throughout the average year at these mooring sites except at Stratus, where pH fell below 8 half the year from December through May (Fig. 2). Moored observations were consistent with seasonal means from ship-based time series observations at the WHOTS and Iceland sites (Bates et al., 2014; Ólafsson et al., 2009; Ólafsson et al., in preparation).

The seasonal cycle of surface ocean $\Omega_{arag}$ and pH were not always consistent with one another. Seawater $\Omega_{arag}$ is largely determined by variations in the concentration of the carbonate ion ($CO_3^{2-}$); pH is influenced by gas exchange of $CO_2$, physical conditions, and biological activity. Observations of surface ocean pH were consistent with a seasonal thermodynamic response, i.e., pH decrease (increase) with SST increase (decrease), at the four subtropical open ocean sites and at the Papa mooring (Figs. 2–4). However,

this strong relationship was not consistent at the subarctic Iceland site. At this site, pH and SST were positively correlated (Fig. 4), suggesting the seasonality of surface ocean pH was dominated by biological activity in the summer and/or winter mixing of upwelled deep water low in temperature and pH (Chen et al., 2007; Takahashi et al., 1993). At all open ocean sites, $\Omega_{arag}$ was highest during summer months, which led to the timing of low $\Omega_{arag}$ and low pH conditions to be anticorrelated over the annual cycle at all open ocean sites except Iceland (Figs. 2–4). This pattern at the Iceland mooring was consistent with seasonality of

surface $\Omega_{arag}$ and pH derived from quarterly ship-based observations at the same site (Bates et al., 2014; Ólafsson et al., in preparation).

Comparisons to pre-industrial bounds of variability also revealed differences between open ocean sites. All open oceans sites experienced surface $\Omega_{arag}$ outside the bounds of pre-industrial variability year-round with the exception of BOBOA and Iceland.

Present-day surface $\Omega_{arag}$ conditions still partially overlapped with pre-industrial conditions at BOBOA during the monsoon season from June through August (Fig. 3) and at Iceland during the summer to fall transition in August and September (Fig. 4). However, present-day surface pH observations fall completely outside pre-industrial pH conditions at all open ocean sites year-round, except at BOBOA where there was a slight overlap of 4% in August (Figs. 2–4).

The coastal mooring sites experienced higher subseasonal to seasonal variability in surface pH and $\Omega_{arag}$ compared to the open ocean sites (Figs. 5 and 6). Each coastal time series exhibited clear seasonal patterns with annual amplitudes of $\Omega_{arag}$ ranging from 0.66 to 1.32 (Table 4). Gray's Reef and Chá bă experienced the highest subseasonal to seasonal variability in surface pH and $\Omega_{arag}$, likely driven by upwelling/relaxation/downwelling dynamics that can change rapidly at Chá bă (Alford et al., 2012; Hickey and Banas, 2003) and high productivity and freshwater inputs in the spring and fall at Gray's Reef (Reimer et al., in review; Salisbury

et al., 2009; Xue et al., 2016). We also observed the lowest pH values (7.8) and surface $\Omega_{arag}$ values close to undersaturation ($\Omega_{arag}<1$) primarily in the winter at Chá bă and in the spring at CCE2 (Fig. 5). These observations of near-undersaturated conditions are consistent with other observations and models within the northern CCE where the Chá bă mooring resides (Harris et al., 2013; Hauri et al., 2013) and may indicate respiration in the absence of photosynthetic uptake typical of winter/non-bloom periods.





Unlike the subtropical open ocean mooring sites, seasonality of surface ocean pH at these coastal sites showed strong influence of factors other than SST and were not always correlated with $\Omega_{arag}$ values. At the moorings in the CCE, these parameters (i.e., SST, pH, and $\Omega_{arag}$) generally followed similar seasonal patterns, suggesting factors other than seasonal thermodynamic response influenced surface ocean pH (Fig. 5). However, surface ocean pH and $\Omega_{arag}$ did not always follow the same seasonal pattern at the
Gray's Reef and Gulf of Maine mooring sites (Fig. 6). While SST likely influenced some of the seasonal variation in pH at these sites, biological activity and freshwater input also influenced seasonality of the carbonate system at these U.S. East Coast locations (Reimer et al., in review; Salisbury et al., 2009; Xue et al., 2016).

In general, the coastal sites experienced $\Omega_{arag}$ outside of pre-industrial range mainly during winter. One exception was Chá bǎ,
where the bottom of the pre-industrial envelope of variability was positioned within a broadly distributed IQR (i.e., the box within the boxplots of Fig. 4) of the observations. Small changes in monthly IQR during spring through fall caused high month-to-month variability in the overlap with pre-industrial conditions, indicating this system is on the threshold of a shift outside pre-industrial conditions during this time of the year. Observations of pH at Chá bǎ followed this same pattern. In general, present-day observations of pH fell outside pre-industrial conditions more so than $\Omega_{arag}$ at all coastal sites (Figs. 5 and 6).

Finally, similar to the coastal moorings, the coral reef mooring sites also experienced subseasonal to seasonal variability but not as large as within the subtropical coastal systems (Fig. 7). Mean annual $\Omega_{arag}$ at the Caribbean (La Parguera) and Pacific (Chuuk) moorings was 3.62 and 3.42, respectively, while mean pH was 8.02 and 8.01, respectively (Fig. 7, Table 4). With the exception of low $\Omega_{arag}$ outliers at Chuuk, most $\Omega_{arag}$ conditions were >3 throughout the year at both sites, and surface pH observations were
>7.9. The seasonal cycle of pH and $\Omega_{arag}$ was more pronounced at La Parguera with relatively consistent monthly range in variability, but the Chuuk site experienced greater variability December through April, likely driven by local mixing during the trade winds season. Even with small seasonal fluctuations in tropical ocean temperature, both coral mooring sites did show patterns of pH and $\Omega_{arag}$ seasonality associated with SST, with lower pH and $\Omega_{arag}$ values coinciding with slightly warmer summer months and higher pH and $\Omega_{arag}$ values during winter (Fig. 7). Present-day variability at these sites did not cause extensive overlap with
pre-industrial conditions. Present-day surface pH observations fell completely outside pre-industrial conditions year-round at both coral reef sites (Fig. 7). Present-day $\Omega_{arag}$ conditions at La Parguera were largely outside of pre-industrial bounds year round, while this mainly occurred during the season of lowest variability from May to November at Chuuk (Fig. 7).

The results from these 12 mooring time series highlight the different patterns of variability of surface ocean $\Omega_{arag}$ and pH in both
space and time. Figure 8 compares the relative influence of subseasonal, seasonal, and interannual variability at the mooring locations. Since the mooring observations were well distributed throughout the year, we are confident in the subseasonal and seasonal estimates of variability. However, considering that most of the time series were only 2 to 5 years long, we expect to refine the estimates of interannual variability as we obtain more observations over the coming years. For example, ENSO is a driver of ocean conditions, including biogeochemistry, at CCE2 (Nam et al., 2011). While there were weak El Niño-like conditions that
developed in the tropical Pacific in 2014 (McPhaden, 2015), there were no major La Niña or El Niño anomalies during the CCE2 time series used in this analysis (March 2012–March 2015). Hence, the estimate of interannual variability presented here is likely an underestimate of the true interannual signal at this location. In addition, this was a period of anomalously rapid warming in the Gulf of Maine, which may have caused $\Omega_{arag}$ to trend higher due the reduced solubility of $\Omega_{arag}$ in warmer waters (Mills et al., 2013; Pershing et al., 2015). Potenial variations in warming trends over time would also impact interannual variabilty of $\Omega_{arag}$
observations in the Gulf of Maine as the time series continues.



The coastal sites generally experienced higher subseasonal to interannual $\Omega_{arag}$ variability than the open ocean and coral reef sites. Relative to other modes of variability, interannual $\Omega_{arag}$ variability tended to be low at all sites except for at Chá bă, Gray's Reef, and CCE2 (Fig. 8a). The other sites tended to be equally influenced by subseasonal and seasonal variability with the exception of the Iceland mooring site, which was controlled more by seasonal variability over the annual cycle (Fig. 8a); however, subseasonal variability played a large role in August through October (Fig. 4). For pH, most mooring sites exhibited similar modes of variability with low interannual variability and approximately equal influence from seasonal and subseasonal variability (Fig. 8b). Similar to $\Omega_{arag}$, Chá bă, Gray's Reef, and CCE2 were the clear outliers with the highest values of interannual pH variability.

### 3.2 Biologically relevant $\Omega_{arag}$ thresholds

Research on response of shellfish larvae living in nearshore environments of the CCE and Gulf of Maine to changes in carbonate chemistry allowed us to investigate when observations at the Chá bă, CCE2, and Gulf of Maine moorings exceeded biological thresholds. *Crassostrea gigas*, the Pacific oyster whose larvae are raised in hatcheries in coastal Washington and Oregon, has shown sublethal impacts on larval development, such as shell development and growth, when exposed to levels of $\Omega_{arag} < 2.0$ (Barton et al., 2012) and acute impacts when $\Omega_{arag} < 1.5$ (Waldbusser et al., 2015a,b). Other studies suggest that chronic exposure thresholds for the larvae of *Ostrea lurida*, the Olympia oyster, and *Mytilus californianus*, the California mussel, occur at $\Omega_{arag} < 1.4$ (Hettinger et al., 2013) and $\Omega_{arag} < 1.8$ (Gaylord et al., 2011), respectively. All of these shellfish larvae, whether naturally occurring or hatchery raised, are found in coastal environments in the region of the Chá bă mooring and *M. californianus* also exist farther south in the nearshore region of the CCE2 mooring. In addition, larvae of *Mya arenaria*, the soft-shell clam commercially harvested on tidal mudflats of the western Gulf of Maine, has shown lack of shell formation and growth in laboratory experiments at $\Omega_{arag} < 1.6$ (Salisbury et al., 2008).

Monthly climatology of $\Omega_{arag}$ developed from the mooring observations at Chá bă suggest that present-day $\Omega_{arag}$ conditions reached chronic exposure levels for *C. gigas* larvae ($\Omega_{arag} < 2.0$) over 50% of the time from November to March, with nearly the entire months of December through March at $\Omega_{arag}$ values less than 2.0 (Fig. 9b). These present-day conditions prevailed over more of the year compared to pre-industrial times, when the most extensive chronic exposure occurred only up to 64% during March (Fig. 9a). Conditions that cause acute responses in *C. gigas* larvae ($\Omega_{arag} < 1.5$) were miminal year-round at Chá bă except for March, when these conditions persisted in the present day during 37% of the month (Fig. 9b) and only 14% of the month during the pre-industrial (Fig. 9a). A similar seasonal pattern also existed for *O. lurida* larvae ($\Omega_{arag} < 1.4$), when chronic exposure levels in March exceeded 27% during the present (Fig. 9b) compared to only 11% during pre-industrial (Fig. 9a). For *M. californianus* larvae, present-day chronic exposure levels ($\Omega_{arag} < 1.8$) prevailed over 40% of the time in January through March at Chá bă while there was less chronic exposure at CCE2, at 11 to 38% of time in March through July (Fig. 9b). In both cases, present-day exceedance of these thresholds prevailed over fewer months and at a smaller percentage of the time during those months (Fig. 9a). For *M. arenaria*, present-day $\Omega_{arag}$ conditions exceeded chronic exposure levels at the Gulf of Maine mooring between 11 to 31% of the time during December through April, with peak exposure levels in February and March (Fig. 9b). In contrast to the CCE, which experienced corrosive $\Omega_{arag}$ conditions before ocean acidification, Gulf of Maine surface water conditions did not exceed biological thresholds for *M. arenaria* at any point during the year in pre-industrial times (Fig. 9a).

These observations suggest that present-day coastal $\Omega_{arag}$ conditions exceeded thresholds for sublethal effects on shellfish larvae in the Gulf of Maine and during both present-day and pre-industrial times at Chá bă and CCE2. However, present-day coastal





conditions surpass these thresholds more often than pre-industrial times (Fig. 9). In some cases, unfavorable surface ocean $\Omega_{arag}$ conditions overlap with the spawning season. Coastal conditions of $\Omega_{arag}$ <1.4 at Chá bă do not currently occur during the May to August *O. lurida* larvae spawning season. *M. californianus* tends to spawn year-round, and while natural populations of *C. gigas* do exist in Washington coastal waters and tend to spawn in the late summer, hatcheries raise *C. gigas* larvae year-round. Mooring

observations suggest that present-day chronic exposure effects on *M. californianus* larvae may be more common in the winter in the northern CCE and in the spring in the southern CCE (Fig. 9). The summer spawning season of natural populations of *C. gigas* avoids chronic and acute exposure levels during winter months; however, hatcheries may encounter these conditions if raising larvae during this time. In the Gulf of Maine, *M. arenaria* spawns when seawater temperatures reach 10°C, which during the moored time series occurred in May through November (Fig. 6). According to the Gulf of Maine mooring observations through

2013, corrosive conditions of $\Omega_{arag}$ <1.6 did not occur during this spawning season (Fig. 9b). However, maximum SST observations in April of 9.7°C were at the verge of this spawning threshold, and rapid warming in the Gulf of Maine of 0.23°C yr$^{-1}$ since 2004 suggest SST as of April 2015 may have exceeded 10°C at the mooring site (Mills et al., 2013; Pershing et al., 2015). If this warming causes *M. arenaria* to begin spawning in April, larvae may become exposed to $\Omega_{arag}$ conditions that limit shell formation and growth (Fig. 9b).

While these observations on the continental shelf were offshore from the inshore habitats where natural populations of shellfish and oyster hatcheries exist, these results provide valuable information on endmember coastal conditions that affect the nearshore regions. These monthly climatologies suggest surface water conditions corrosive to shellfish larvae presently exist year-round in the CCE (primarily during winter/spring) and during winter/spring in the Gulf of Maine. For shellfish hatcheries that utilize real-

20 time coastal ocean acidification data and monitor conditions within their facilities, managing the impacts of these corrosive conditions on larvae may be possible. These climatologies may also inform the development of experiments testing the vulnerability of shelled organisms in other coastal regions. For example, target species may include ecologically or economically important species that undergo critical life stages when low $\Omega_{arag}$ conditions persist during spring in the region around Gray's Reef (Fig. 6). However, the coastal mooring climatologies also illustrate that low $\Omega_{arag}$ and low pH conditions do not always coincide

in the natural environment, and experiments testing how $\Omega_{arag}$, pH, and other stressors independently affect marine organisms are necessary for understanding ocean acidification impacts under the diversity of present-day conditions (Breitburg et al., 2015).

### 3.3 Comparison to models and ship-based data syntheses

Since high-frequency autonomous ocean acidification time series are relatively new, much of our current knowledge about ocean carbonate variability comes from ship-based observations. Due to the limitations of ship-based oceanography, these observations

can have a seasonal measurement bias leading to errors in seasonal climatology estimates and only capture opportunistic stochastic events, which has resulted in limited knowledge about the influence of subseasonal processes on ocean carbonate variability. In general, we found fairly good agreement between annual mean mooring $\Omega_{arag}$ observations and annual mean ship-based data syntheses, which primarily used repeat hydrographic cruise data from the Global Data Analysis Project (Key et al., 2004). Both the Jiang et al. (2015) and Takahashi et al. (2014) data syntheses overestimated surface ocean $\Omega_{arag}$ at the Stratus mooring in the

South Pacific by 0.31 (Fig. 1; Table 4). Undersampling likely contributed to this discrepancy. Moored observations revealed the lowest $\Omega_{arag}$ conditions during August through October; however, ship-based observations were lacking in this region of the Southern Hemisphere during this season. Annual mean surface $\Omega_{arag}$ at the four U.S. coastal mooring sites tended to reflect mean open ocean conditions characterized by the ship-based data synthesis presented in Figure 1; however, direct observations in the two coral reef environments suggest open ocean carbonate chemistry was modified on the reefs and in these two cases, resulted in





reduced annual mean $\Omega_{arag}$ compared to the ship-based data syntheses (Fig. 1; Table 4). The ship-based data syntheses also slightly overestimated annual mean $\Omega_{arag}$ at WHOTS, KEO, BOBOA, Iceland, and Chá bă; however, these overestimations were roughly within the change in $\Omega_{arag}$ expected between the time of the mooring observations (typically 2010–2015) and the ship-based observations (adjusted to a reference year of 2000 by Jiang et al. [2015] and 2005 by Takahashi et al. [2014]). Assuming a global

average rate of change of surface ocean $\Omega_{arag}$ of -0.008 $yr^{-1}$ (Bates et al., 2014; Takahashi et al., 2014), the change over this 5–15 year period would be 0.04–0.12.

Of the 10 mooring locations with observations presented in the Takahashi et al. (2014) data synthesis, seasonal variability was overestimated by the ship-based observations at all open ocean sites except Iceland, and underestimated at Iceland and the coastal

and coral reef sites (Table 4). These differences could be driven by sparse ship-based data in space and time used to estimate climatological seasonal variability in the Takahashi et al. (2014) synthesis. This analysis demonstrates that in addition to new information about subseasonal variability that is not captured by ship-based observations, moored observations can also be used to improve ship-based data synthesis estimates of seasonal to annual $\Omega_{arag}$ conditions in undersampled regions such as the Southern Hemisphere, Iceland Sea, and coastal systems.

Overall, earth system models tend to underestimate natural variability of the carbonate system. The series of earth system models used by Friedrich et al. (2012) underestimated both seasonal and interannual variability of surface $\Omega_{arag}$ at all mooring locations except for WHOTS and Stratus, which were the sites with the lowest variability (Table 4). These underestimations are expected at the coastal and coral sites since the models do not capture small-scale biogeochemical processes occurring in these environments.

When Friedrich et al. (2012) extrapolated an average annual $\Omega_{arag}$ amplitude of ~0.1 in subtropical oligotrophic open ocean regions to coral locations, they concluded that present-day coral conditions fell 5 times outside the pre-industrial range of variability. However, we found that actual seasonal variability was 2 to 3 times higher than 0.1 at the Chuuk and La Parguera mooring locations (Table 4), and present-day $\Omega_{arag}$ conditions were only 1 to 2 times below the pre-industrial range of variability (Fig. 7). On the other hand, we found CCSM3-based estimates of pre-industrial envelop exceedance by Cooley et al. (2009) to be conservative in

some regions. They found that by 2050 all regions will experience surface $\Omega_{arag}$ conditions outside pre-industrial bounds of variability with emphasis in low-latitude regions. Our present-day mooring observations suggest that not only has the shift outside of pre-industrial conditions already occurred year-round at the low latitude coral reef sites but also at subtropical and subarctic open ocean sites.

Unlike global earth system models, some regional models are able to resolve small-scale coastal processes and may provide better estimates of natural variability in these dynamic systems. We found the highest levels of natural variability at the Chá bă mooring location with an annual range of surface ocean $\Omega_{arag}$ of 3.3, from $\Omega_{arag}$ values of 1.06 to 4.36 (Fig. 5). Estimates of this range for the northern CCE produced by a regional model were only 0.2, which led to the conclusion that surface ocean $\Omega_{arag}$ conditions in 2005 were already outside the bounds of pre-industrial conditions (Hauri et al., 2013). Observations at Chá bă from 2010 to 2014

suggest conditions only fell partially outside pre-industrial variability, primarily during the lower-variability season from October to February (Fig. 5).

## 4 Conclusions

Direct, high-resolution observations of seawater $p(CO_2)$ and pH reveal that marine life are currently exposed to surface ocean pH and $\Omega_{arag}$ values outside the envelope of pre-industrial variability they have adapted to at all 12 study locations. Marine life at



several study locations are also exposed to conditions exceeding thresholds that may impact growth and survival of shellfish and conditions approaching undersaturation ($\Omega_{arag}<1$). These ocean acidification impacts are occurring at the same time that marine organisms are also experiencing different modes of temporal variability and other anthropogenic stressors, which can be unique to distinct locations and seasons. This high-resolution mooring work provides a new perspective on variability, since earth system

models and ship-based observations generally underestimate the temporal variability of surface ocean $\Omega_{arag}$ conditions. These results highlight the need to further interrogate these biases, which are often the basis for predictions of future ocean acidification impact. In most cases, such as the WHOTS mooring time series, ocean carbonate observations are also paired with additional autonomous physical and biogeochemical measurements at the surface and at depth, as well as long-term ship-based time series measurements, which are not as temporally resolved as the moored measurements but often include biogeochemical parameters

that cannot be measured autonomously. Further synthesis of these data sets from multiple platforms will contribute to improving understanding of the biogeochemical processes controlling carbonate chemistry at these time series locations and to developing parameterizations for global and regional models. Here we focused on assessing ocean carbonate variability and change as a step in advancing these efforts; however, future research at these and other ocean acidification mooring sites should also include assessments of additional biogeochemical parameters, such as dissolved oxygen and optical properties as these new observational

data sets become available.

Sustained, autonomous observations resolving sub-seasonal conditions are likely to match timescales relevant to biological processes in the natural environment such as energy availability, biological threshold exceedance, seasonal spawning, and recruitment. This characterization of temporal variability of ocean carbonate is one of the major challenges to understanding how

anthropogenic change will impact marine life. Impacts to marine life could manifest through one or a combination of the following environmental stressors: the slow, steady change over time as pH and $\Omega_{arag}$ conditions respond to gradual ocean uptake of anthropogenic $CO_2$; the point in time when pH and $\Omega_{arag}$ conditions leave the pre-industrial envelope of variability to which organisms have adapted; when average or seasonal pH and $\Omega_{arag}$ conditions pass a certain threshold; or when episodic corrosive conditions surpass a tipping point in terms of frequency and duration. Our mooring time series exhibit periods of time when surface

ocean pH and $\Omega_{arag}$ conditions fall outside pre-industrial bounds of variability along with surpassing biologically relevant thresholds, but also time periods where none or only one of these stressors is present. A broad understanding of how this myriad of environmental stressors impact marine life will require a range of approaches including continued and expanded biogeochemical observing, research interrogating the fundamental processes underlying organism response to ocean acidification under laboratory and field conditions, experiments designed to address how different modes of variability and change impact organisms, and paired

chemical and biological observations in the field to assess potential present-day impacts. Characterizing natural variability and biological impact of ocean acidification conditions at key locations will also be fundamental to improving vulnerability assessments seeking to quantify the economic impact of ocean acidification at local to global scales.

### Acknowledgments

The $CO_2$ and ocean acidification observations were funded by NOAA's Climate Observation Division (COD) in the Climate

Program Office and NOAA's Ocean Acidification Program. The maintenance of the Stratus and WHOTS Ocean Reference Stations were also supported by NOAA COD (NA09OAR4320129). Additional support for buoy equipment, maintenance, and/or ancillary measurements was provided by NOAA through the U.S. Integrated Ocean Observing System office: for the La Parguera buoy under a Cooperative Agreement (NA11NOS0120035) with the Caribbean Coastal Ocean Observing System, for the Chá bã buoy under a Cooperative Agreement (NA11NOS0120036) with the Northwest Association of Networked Ocean Observing





System, for the Gray's Reef buoy under a Cooperative Agreement (NA11NOS0120033) with the Southeast Coastal Ocean Observing Regional Association, and for the Gulf of Main buoy under a Cooperative Agreement (NA11NOS0120034) with the Northeastern Regional Association of Coastal and Ocean Observing Systems. This $CO_2$ and ocean acidification observation network would not be possible without the diligent efforts of PMEL technical and engineering staff, as well as current and former

5    partners and their staff that support the maintenance of the buoys including: technical staff of the Upper Ocean Processes Group at the Woods Hole Oceanographic Institution and the crews of the NOAA and UNOLS vessels used for maintenance (WHOTS and Stratus); Jón Ólafsson and Héðinn Valdimarsson (Iceland); Chris O'Brien, Rudi Hermes, and Dave Zimmerman (BOBOA); John Mickett (Chá bǎ); Mark Ohman, Paul Chua, and David Glassier (CCE2); Yongchen Wang and Scott Noakes (Gray's Reef); Chris Hunt and Shawn Shellito (Gulf of Maine); Melissa Melendez Oyola (La Parguera); and Charity Lee and Seong Kim (Chuuk).

10   A special thank you to Stacy Maenner-Jones, Randy Bott, Sylvia Musielewicz, John Osborne, and Colin Dietrich of the PMEL $CO_2$ mooring team, who support all aspects of data collection and quality control, and also to Li-Qing Jiang who provided the base map for Figure 1. PMEL contribution 4435 and JISAO contribution 2506.

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




**Table 1**. Details on each surface seawater $p(CO_2)$ and pH mooring time series including abbreviation, name, coordinates, and dates of time series. n is the total number of 3-hourly samples for each pH and $\Omega_{arag}$ time series. Finalized data sets can be found organized by mooring location by navigating through the clickable $CO_2$ data portal map at http://cdiac.ornl.gov/oceans/Moorings/.

| Abbreviation | Full Name | Latitude | Longitude | Dates of time series | n |
|---|---|---|---|---|---|
| WHOTS | WHOI Hawaii Ocean Timeseries Station | 22.7 | -158.0 | Dec 2004 - Mar 2014[a] | 18532 |
| Stratus | Stratus | -19.7 | -85.6 | May 2012 - Jul 2013 | 3193 |
| BOBOA | Bay of Bengal Ocean Acidification Mooring[b] | 15.0 | 90.0 | Nov 2013 - Aug 2015 | 4085 |
| Iceland | North Atlantic Ocean Acidification Mooring | 68.0 | -12.6 | Aug 2013 - Jul 2015 | 2671 |
| Papa | Ocean Station Papa | 50.1 | -144.8 | Jun 2010 - Dec 2013 | 6005 |
| KEO | Kuroshio Extension Observatory | 32.3 | 144.6 | Nov 2011 - Jun 2015 | 5540 |
| Chá bă | Chá bă Mooring within the Northwest Enhanced Moored Observatory | 48.0 | -125.0 | Jul 2010 - Apr 2014 | 5115 |
| CCE2 | California Current Ecosystem 2 | 34.3 | -120.8 | Mar 2012 - Mar 2015 | 3197 |
| Gray's Reef | National Data Buoy Center buoy at Gray's Reef National Marine Sanctuary | 31.4 | -80.9 | Nov 2010 - Sep 2015 | 4184 |
| Gulf of Maine | Coastal Western Gulf of Maine Mooring | 43.0 | -70.5 | Sep 2010 - Jul 2013 | 5392 |
| Chuuk | Chuuk Lagoon Mooring | 7.5 | 151.9 | Dec 2011 - Sep 2014 | 2274 |
| La Parguera | La Parguera Ocean Acidification Mooring | 18.0 | -67.1 | Dec 2012 - Oct 2014 | 2833 |

Notes: [a] The WHOTS time series includes mooring $p(CO_2)$ observations from the Multi-disciplinary Ocean Sensors for Environmental Analyses and Networks station at 22.8°N, 158.1°W from 2004-2007. [b] The BOBOA mooring is embedded within the Research Moored Array for African–Asian–Australian Monsoon Analysis and Prediction (RAMA; McPhaden et al., 2009).





**Table 2**. Mean and variability, as measured by 1 SD, for $A_T$ and $\Omega_{arag}$ data sets at the WHOTS and La Parguera mooring sites. Discrete measurements for WHOTS are from the Hawaii Ocean Time series program (http://hahana.soest.hawaii.edu/hot/) and for La Parguera are from the Atlantic Ocean Acidification Test-bed project. Proxy $A_T$ measurements are estimated from moored SST and SSS using Lee et al. (2006) $A_T$-SSS relationships. $p(CO_2)$ and pH used to calculate $\Omega_{arag}$ are direct observations from the mooring time series; $A_T$ used to calculate $\Omega_{arag}$ are the $A_T$ proxy measurements.

|  |  | WHOTS | La Parguera |
|---|---|---|---|
| *mean* | discrete $A_T$ | 2307 | 2290 |
|  | proxy $A_T$ | 2310 | 2298 |
| *SD* | discrete $A_T$ | 16 | 44 |
|  | proxy $A_T$ | 9 | 39 |
| *mean* | $\Omega_{arag:\ discrete}$ | 3.68 | 3.60 |
|  | $\Omega_{arag:\ pCO2-AT}$ | 3.58 | 3.65 |
|  | $\Omega_{arag:\ pH-AT}$ | 3.56[a] | 3.59 |
|  | $\Omega_{arag:\ pCO2-pH}$ | 3.60[a] | 3.50 |
| *SD* | $\Omega_{arag:\ discrete}$ | 0.11 | 0.16 |
|  | $\Omega_{arag:\ pCO2-AT}$ | 0.07 | 0.12 |
|  | $\Omega_{arag:\ pH-AT}$ | 0.05[a] | 0.13 |
|  | $\Omega_{arag:\ pCO2-pH}$ | 0.08[a] | 0.19 |

Notes: [a] While other observations listed here span the full annual cycle, the WHOTS mooring pH observations only cover the months of June to November.





**Table 3**. Sources of error to the calculation of $\Omega_{arag}$ at SST = 25°C, SSS = 35, $p(CO_2)$ = 370 μatm, pH = 8.1, $A_T$ = 2350, and $\Omega_{arag}$ = 3.7. Total estimated absolute (relative) uncertainty was calculated using the root sum of squares (RSS) method: $RSS = (\sum a^2)^{1/2}$.

| Sources of error | Variable error (±) | Effect on error of $\Omega_{arag}$ calculation ($a$) |
|---|---|---|
| *$pCO_2$ and pH pair:* | | |
| $p\,CO_2$ measurement | 2 μatm[a] | 0.02 |
| pH measurement | 0.018 | 0.32 |
| $K_0$ | 0.004[b] | |
| $K_1$ | 0.015[b] | |
| $K_2$ | 0.03[b] | 0.18[c] |
| *Estimated uncertainty of $\Omega_{arag:\,pCO2\text{-}pH}$:* | | *0.37 (10%)* |
| *$pCO_2$ and $A_T$ pair:* | | |
| $p\,CO_2$ measurement | 2 μatm[a] | 0.02 |
| $A_T$ proxy | 3-20 μmol kg$^{-1}$ [d] | 0.01-0.05 |
| $K_0$ | 0.004[b] | |
| $K_1$ | 0.015[b] | |
| $K_2$ | 0.03[b] | 0.18[c] |
| *Estimated uncertainty of $\Omega_{arag:\,pCO2\text{-}AT}$:* | | *0.19 (5%)* |

Notes: [a] From Sutton et al. (2014b). [b] Error estimates of thermodynamic constants from McLaughlin et al. (2015). [c] Combined effect of thermodynamic constants (Mucci et al. 1983). [d] Range of error from $A_T$ proxies described in methods.



**Table 4**. Descriptive statistics of $\Omega_{arag}$: annual mean, annual amplitude, and 1 SD of annual anomalies from the $CO_2$ and pH mooring observations, from a global data synthesis of ship-based observations (Takahashi et al., 2014), and a biogeochemical model (Friedrich et al., 2012). Dark gray cells represent values larger than observed values; light gray cells represent values lower than observed values. ND signifies no data.

| open ocean sites | annual mean | annual amplitude | SD annual anomalies | coastal and coral reef sites | annual mean | annual amplitude | SD annual anomalies |
|---|---|---|---|---|---|---|---|
| **WHOTS** | | | | **Chá bǎ** | | | |
| observations | 3.49 | 0.17 | 0.05 | observations | 1.88 | 1.32 | 0.45 |
| global synthesis | 3.62 | 0.34 | | global synthesis | 2.06 | 0.67 | |
| model | | 0.25 | 0.09 | model | | ND | ND |
| **Stratus** | | | | **CCE2** | | | |
| observations | 2.67 | 0.13 | 0.07 | observations | 2.53 | 0.76 | 0.31 |
| global synthesis | 2.98 | 0.37 | | global synthesis | ND | ND | |
| model | | 0.35 | 0.11 | model | | ND | ND |
| **BOBOA** | | | | **Gray's Reef** | | | |
| observations | 3.52 | 0.20 | 0.13 | observations | 3.25 | 1.09 | 0.37 |
| global synthesis | 3.59 | 0.24 | | global synthesis | 3.09 | 0.95 | |
| model | | 0.15 | 0.06 | model | | ND | ND |
| **Iceland** | | | | **Gulf of Maine** | | | |
| observations | 1.70 | 0.71 | 0.22 | observations | 1.86 | 0.64 | 0.24 |
| global synthesis | 1.77 | 0.64 | | global synthesis | ND | ND | |
| model | | 0.45 | 0.16 | model | | ND | ND |
| **Papa** | | | | **Chuuk K1** | | | |
| observations | 2.08 | 0.49 | 0.15 | observations | 3.42 | 0.21 | 0.11 |
| global synthesis | 1.83 | 0.55 | | global synthesis | 3.86 | 0.08 | |
| model | | 0.35 | 0.09 | model | | 0.05 | 0.04 |
| **KEO** | | | | **La Parguera** | | | |
| observations | 3.08 | 0.48 | 0.16 | observations | 3.62 | 0.33 | 0.11 |
| global synthesis | 3.32 | 0.61 | | global synthesis | 3.86 | 0.21 | |
| model | | 0.35 | 0.06 | model | | 0.15 | 0.04 |





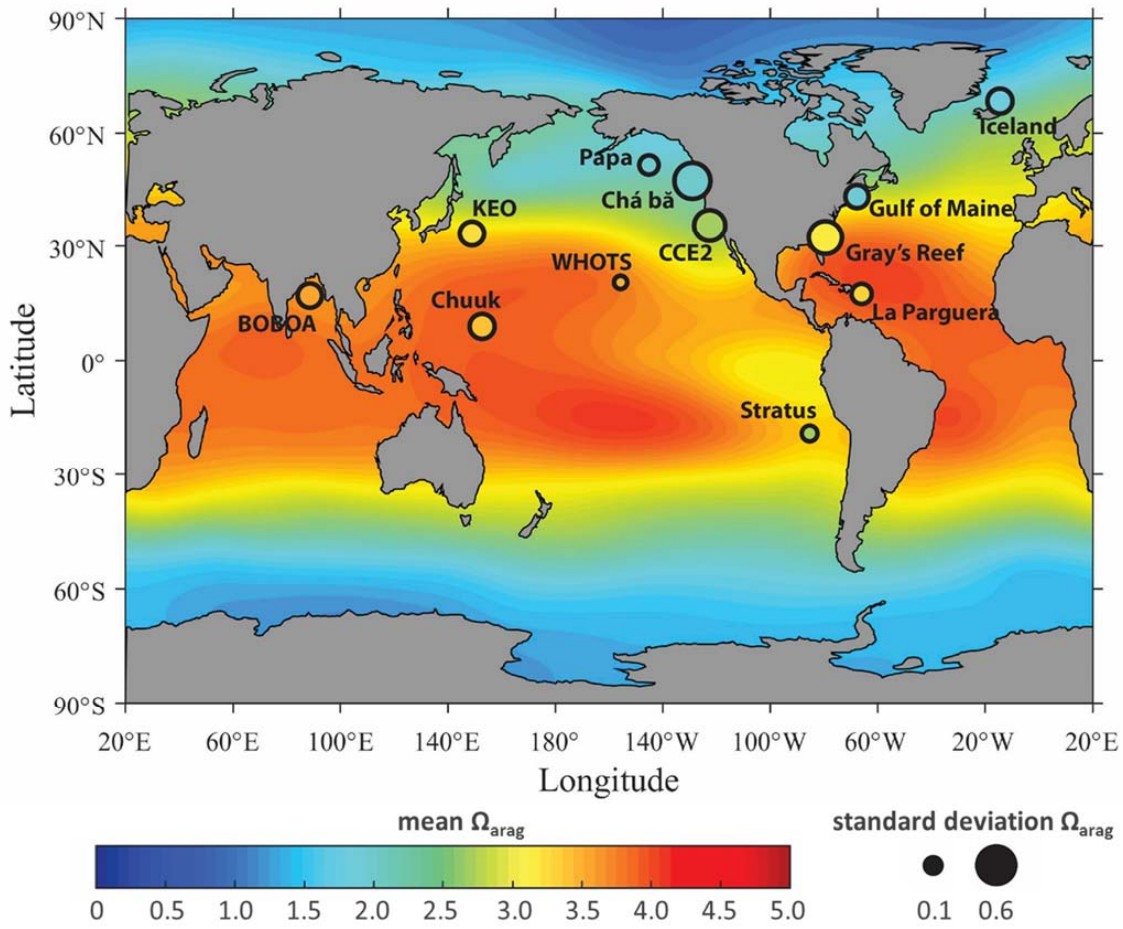

**Figure 1**. Locations of moored $p(CO_2)$ and pH observations. Base map is adapted from Jiang et al. (2015) and shows annual climatological distribution of surface $\Omega_{arag}$ throughout the global oceans. Added to this base map are moorings from this study where symbol color is annual mean $\Omega_{arag}$ (values also listed in Table 4) and size of symbol represents $\Omega_{arag}$ variability as measured by 1 standard deviation (SD) of the annual mean.





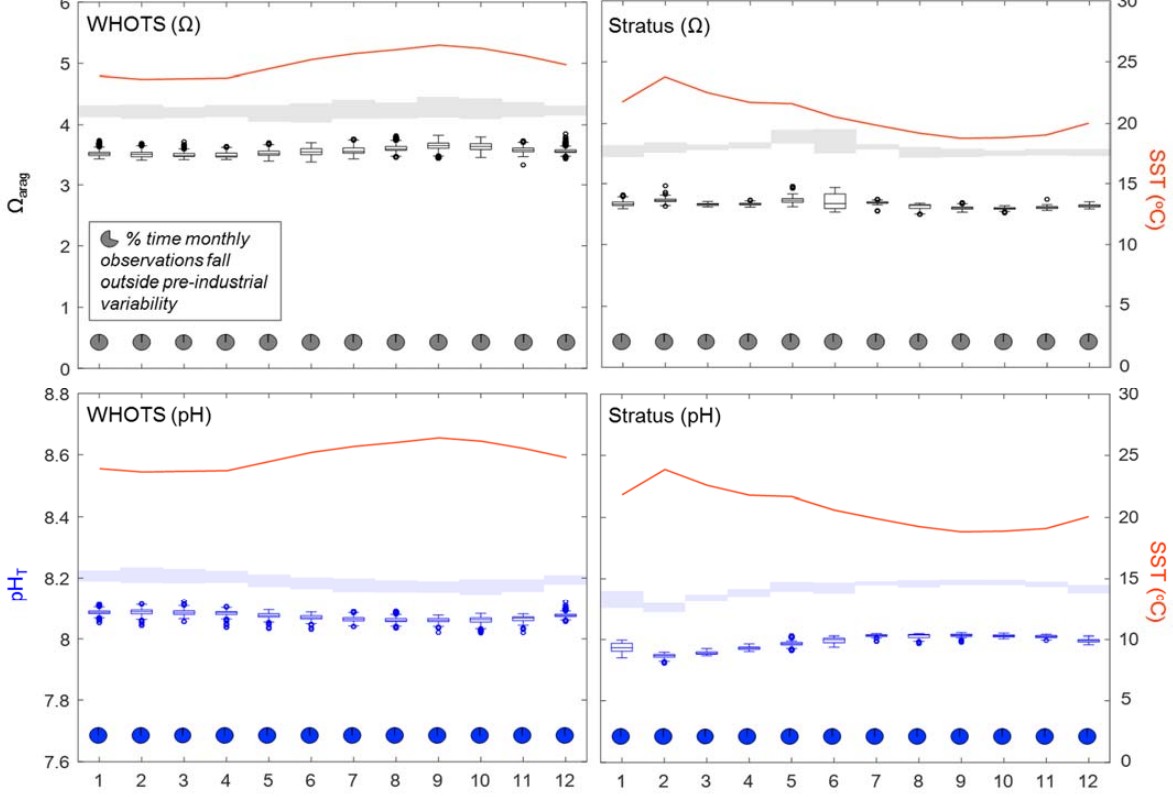

**Figure 2**. Box and whisker plots of present-day monthly surface seawater $\Omega_{arag}$ (top) and pH (bottom) and monthly mean SST (orange lines) at the open ocean mooring locations in subtropical oligotrophic regions (WHOTS, Stratus). Boxes represent data between Q1 and Q3, with the line between representing Q2 (i.e., the median). Whiskers represent 1.5 IQR, or ~2.7 SD; Eqs. (1) and (2), of the upper and lower quartiles with data outside that range shown as outliers (open circles). Outliers here represent natural deviations in ocean chemistry, not measurement outliers, which were removed in the data quality control process. Estimated monthly pre-industrial $\Omega_{arag}$ and pH variability (1.5 IQR or ~2.7 SD) is shown in gray (top) and blue (bottom) shaded areas, respectively. Shaded portion of the pie charts indicate the percent of present-day $\Omega_{arag}$ and pH values falling outside the bounds of pre-industrial variability for each month. For mooring location see Fig. 1 and Table 1.




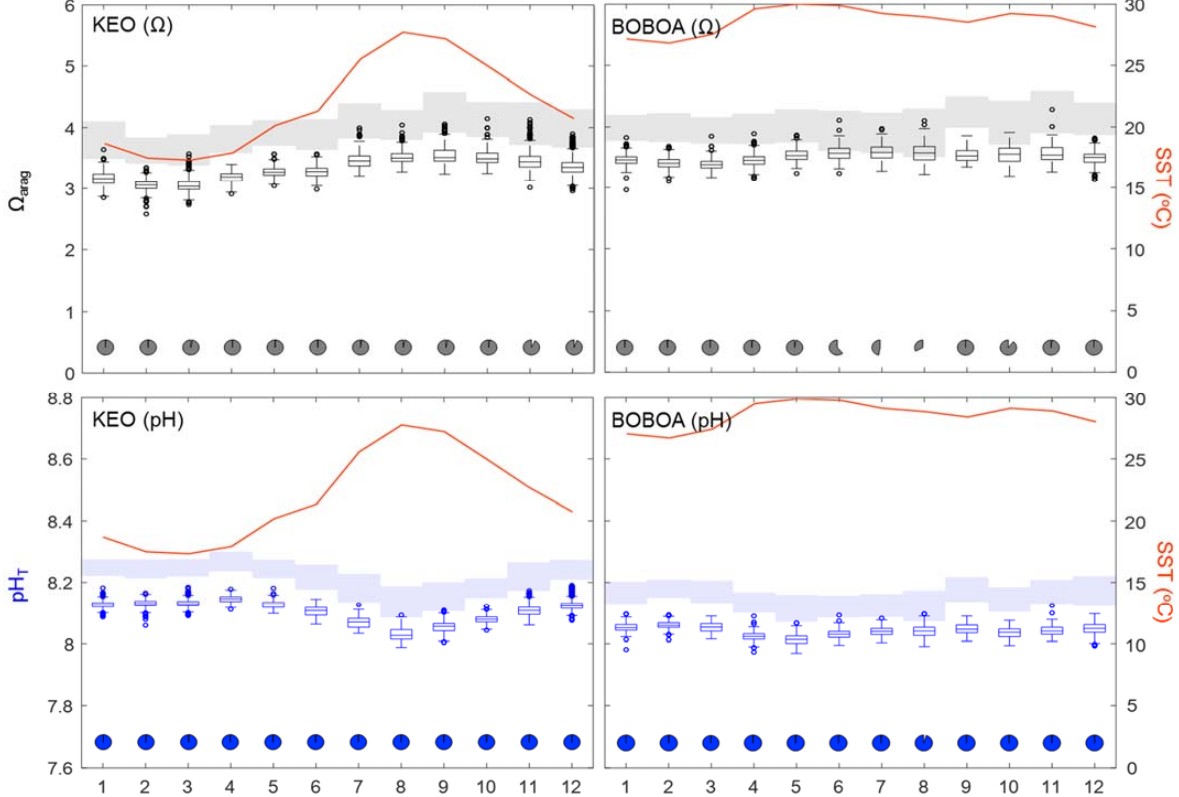

**Figure 3**. Box and whisker plots of present-day monthly surface seawater $\Omega_{arag}$ (top) and pH (bottom) and monthly mean SST at the open ocean mooring locations in biologically productive subtropical regions that experience seasonal monsoons (BOBOA) and tropical cyclones (KEO). See detailed description of figure components in Fig. 2 caption.





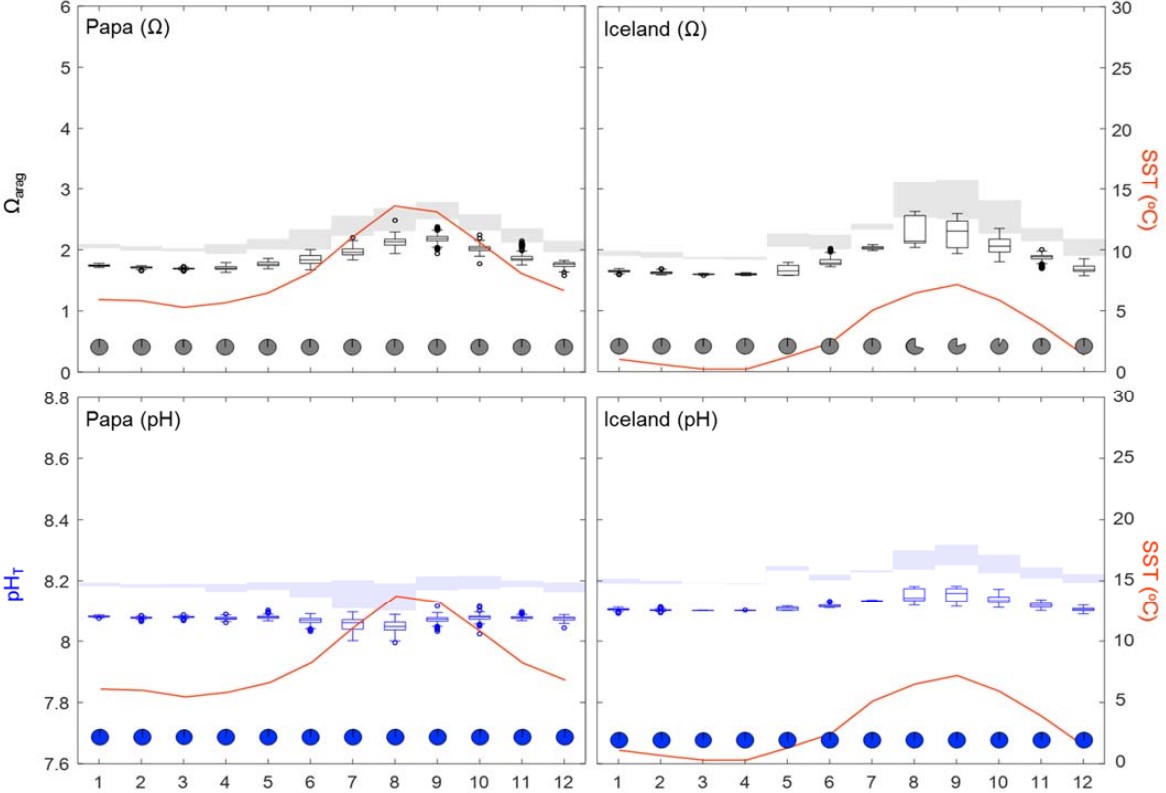

**Figure 4**. Box and whisker plots of present-day monthly surface seawater $\Omega_{arag}$ (top) and pH (bottom) and monthly mean SST at the open ocean mooring locations in subarctic regions with pronounced seasonality of physical and biological conditions (Papa, Iceland). See detailed description of figure components in Fig. 2 caption.





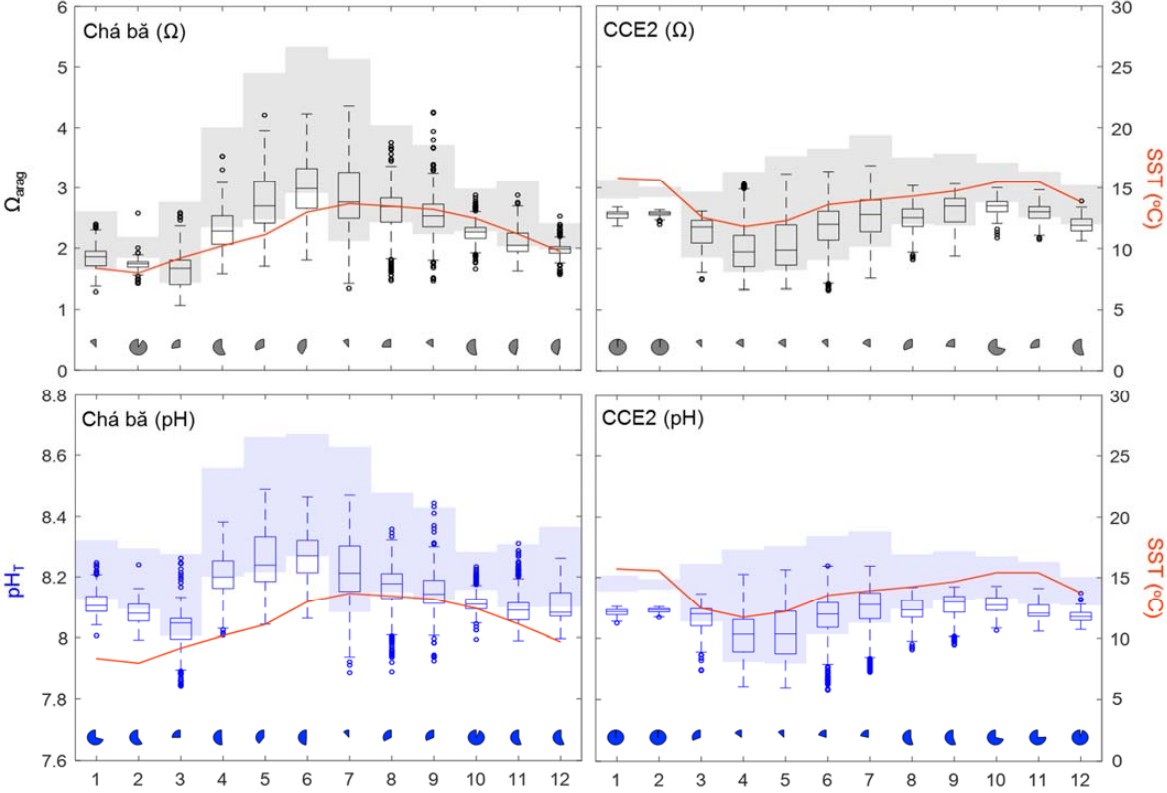

**Figure 5**. Box and whisker plots of present-day monthly surface seawater $\Omega_{arag}$ (top) and pH (bottom) and monthly mean SST at the coastal mooring locations on the continental shelves of the U.S. West Coast (CCE2, Chá bă). See detailed description of figure components in Fig. 2 caption.





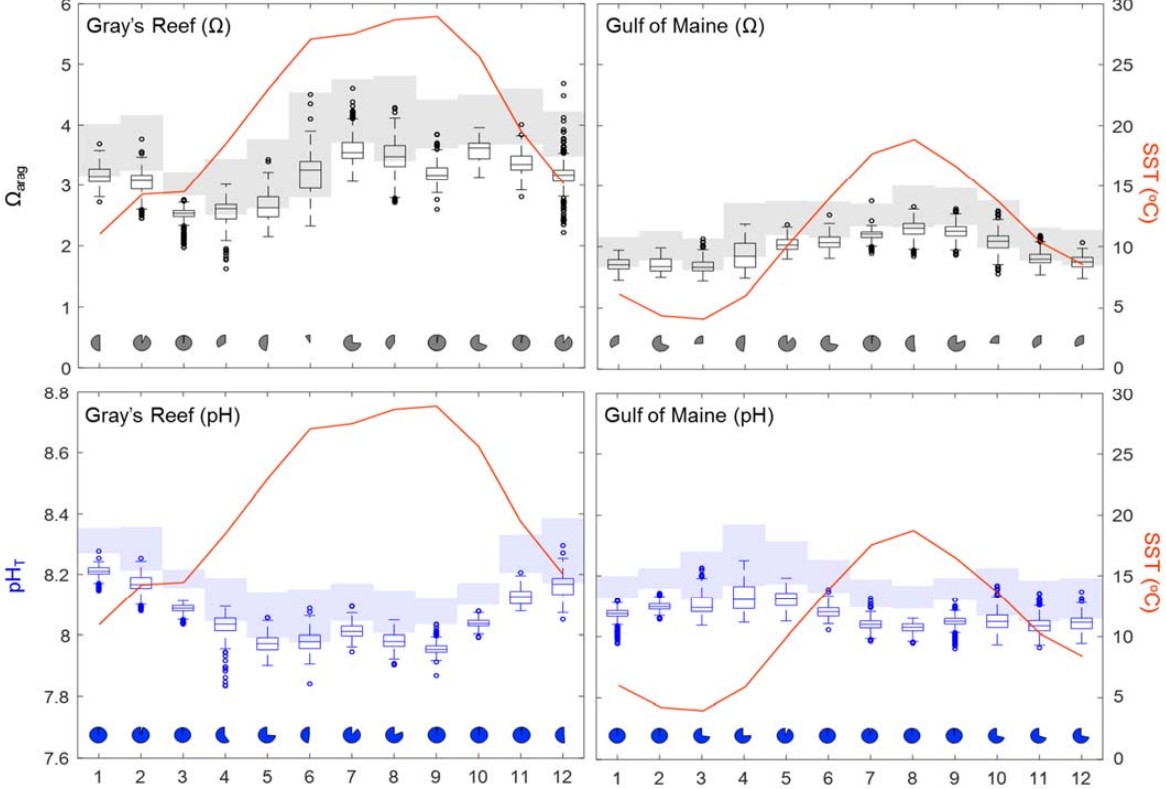

**Figure 6**. Box and whisker plots of present-day monthly surface seawater $\Omega_{arag}$ (top) and pH (bottom) and monthly mean SST at the coastal mooring locations on the continental shelves of the U.S. East Coast (Gulf of Maine, Gray's Reef). See detailed description of figure components in Fig. 2 caption.




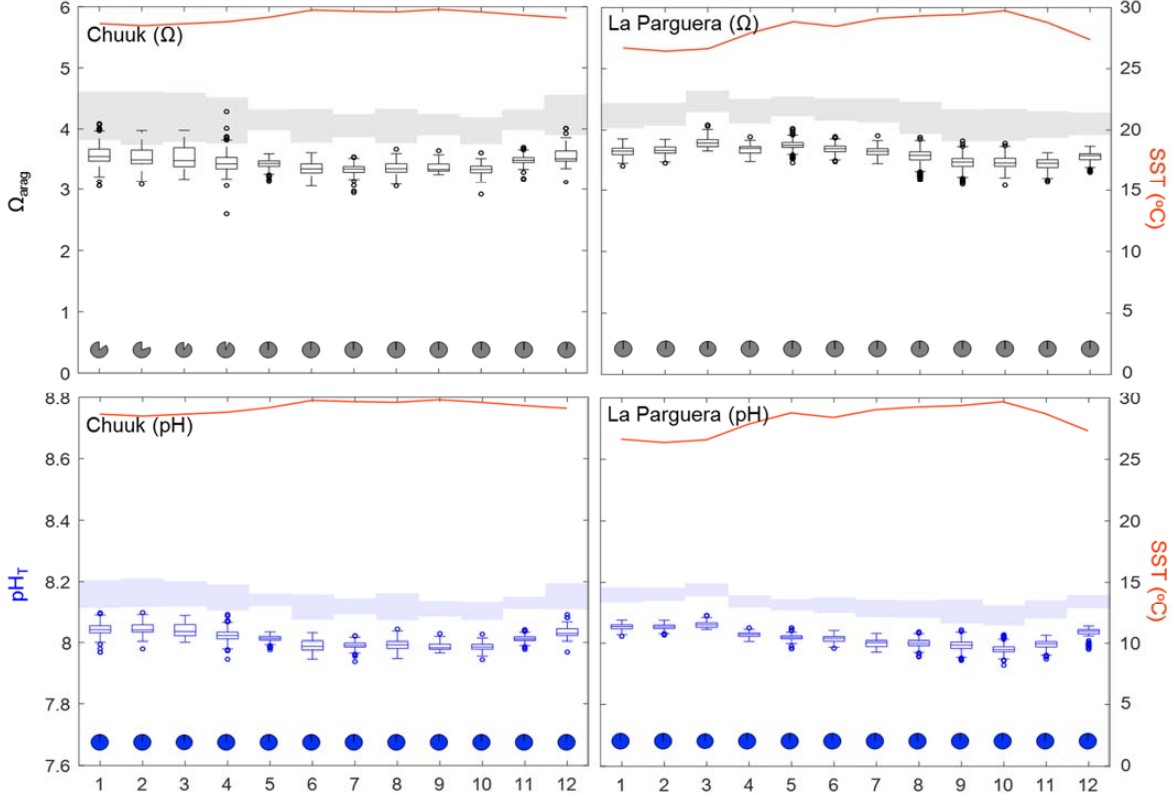

**Figure 7**. Box and whisker plots of present-day monthly surface seawater $\Omega_{arag}$ (top) and pH (bottom) and monthly mean SST in coral reef ecosystems within a semi-closed atoll lagoon in Micronesia (Chuuk) and a patch reef in the Caribbean Sea southwest of Puerto Rico (La Parguera). See detailed description of figure components in Fig. 2 caption.



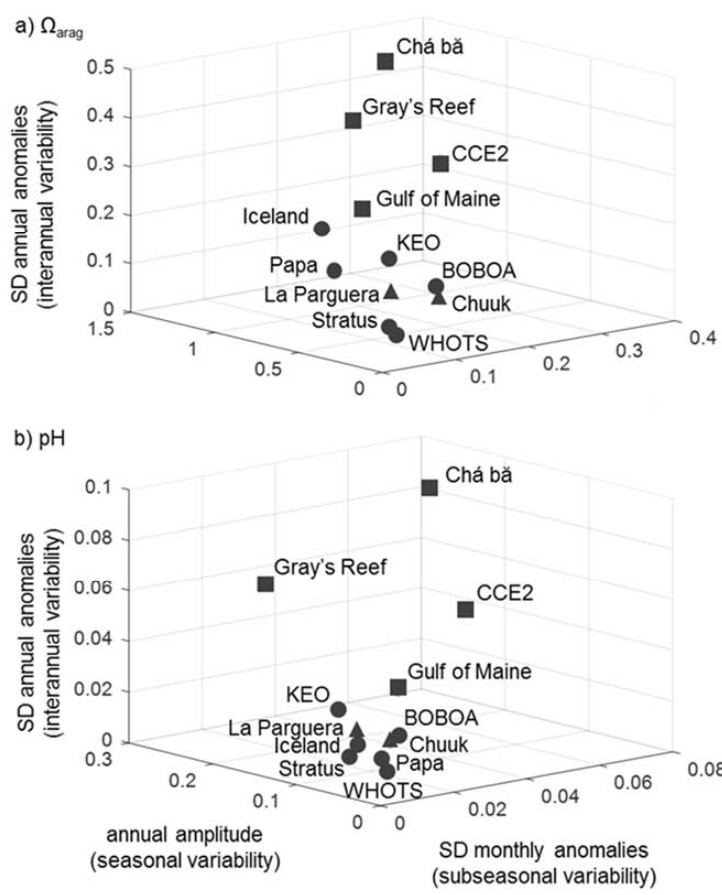

**Figure 8**. Relational plot of different modes of a) $\Omega_{arag}$ and b) pH variability for each ocean acidification mooring location. Statistics describing variability include 1 SD of monthly anomalies (monthly mean – monthly observations), annual amplitude (maximum monthly climatological mean – minimum monthly climatological mean), and 1 SD of annual anomalies (annual mean – mean observations). Circles represent open ocean mooring locations, squares are coastal, and triangles are coral reefs.





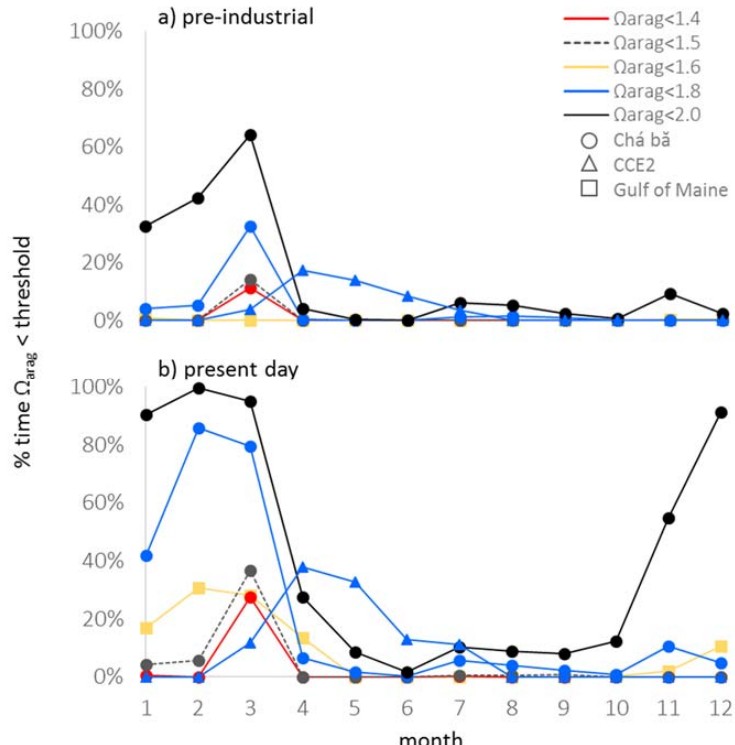

**Figure 9**. Percent time that a) pre-industrial and b) present-day surface seawater $\Omega_{arag}$ conditions fall below biological thresholds: chronic exposure for *Ostrea lurida* larvae at $\Omega_{arag} < 1.4$ in red, acute effect of *Crassostrea gigas* larvae at $\Omega_{arag} < 1.5$ in gray, chronic exposure for *Mya arenaria* larvae at $\Omega_{arag} < 1.6$ in gold, chronic exposure for *Mytilus californianus* larvae at $\Omega_{arag} < 1.8$ in blue, and chronic exposure for *C. gigas* larvae at $\Omega_{arag} < 2.0$ in black. Thresholds at the Chá bă mooring are shown as circles; thresholds at the CCE2 mooring (only for *M. californianus* larvae) are shown as triangles; thresholds at the Gulf of Maine mooring (only for *M. arenaria* larvae) are shown as squares. The one acute threshold is indicated by a dashed line.