# Peer review of "Using present-day observations to detect when anthropogenic change forces surface ocean carbonate chemistry outside pre-industrial bounds"

_Biogeosciences, 2016_

## Referee Comment (RC1) · Anonymous Referee #1 · 26 Apr 2016

The study is an attempt to constrain the variability of surface ocean carbonate chemistry via compiling 3-hourly moored observations for 12 open ocean, coastal, and coral reef locations. Further, these present-day conditions are compared to biologically relevant thresholds associated with ocean acidification.

These are very relevant topics in the context of anthropogenic climate change and definitely within the scope of BG. The paper is detailed and well-written. I would appreciate a more thorough evaluation of state-of-the-art ESMs against this new set of observations, which would be very valuable (as the authors correctly note, ESMs still

have issues in capturing the full magnitude of variability), but that is perhaps beyond the scope of this paper.

Specific comments:
p 1 line 18: Looking at "long-term change" in the context of "natural variability", wouldn't be the challenge the detectability (and correct estimation) of these long-term trends in OA - which then in turn affect marine life?.
p 1 line 29: "Modes" refers to ENSO, NAO etc. - I would recommend the usage of "patterns" or something like that.
p 1 line 35: Although - While?
p 2 line 10: than in open..
p 2 line 19ff: Please also include more recent studies, e.g., Keller at al., 2014 and Rodgers et al., 2015.
p 3 line 15: "and to ground truth carbonate chemistry variability in earth system models." Quite strong wording for what is actually done later, also considering the small number of locations and (partly quite old) models. Rewrite.
p 5 line 1: "or variability" - delete. SD was introduced as variability just one sentence before.
p 5 line 34: Q3 + 1.5 x IQR
p 7 line +/-25: How exactly do you define overlap? At KEO/$\Omega$arag, the whisker in November seems also to be in the gray.
p 8 line 9ff: This paragraph is not really clear, please rephrase.
p 8 line 11: Chá bǎ is shown in Fig. 5, not 4.
p 9 line 2: see p 1 line 29
p 9 line 6: see p 1 line 29
p 12 line 3: see p 1 line 29

Tables & Figures:
The gray and blue features (cells, shades, grids) are hardly visible (printed out, the light gray in Tab. 4 is completely absent, as is the grid in Fig. 8). Please replace with

stronger colors.

—

K. M. Keller, F. Joos, C. C. Raible, Time of emergence of trends in ocean biogeochemistry. Biogeosciences 11, 3647–3659 (2014). doi: 10.5194/bg-11-3647-2014
K. B. Rodgers, J. Lin, T. L. Frölicher, Emergence of multiple ocean ecosystem drivers in a large ensemble suite with an Earth system model. Biogeosciences. 12, 3301–3320 (2015). doi: 10.5194/bg-12-3301-2015

---

## Referee Comment (RC2) · Anonymous Referee #2 · 29 Apr 2016

The authors observed pH and pCO2 of surface seawater for several years by using moor system and calculated present-day monthly omega. Then, they discuss biological thresholds of shellfish using pre-industrial calculated omega. Overall, I agree with the authors. I also would like to know time series changes of pH and omega during observation, but this would be another topic. Since the manuscript is well written, only a few minor comments are attached below.

Fig.2-7 I would like you to add pCO2 data

P4 L1 "Overall uncertainty..." Is it ture ? I think 2 uatm is too small.

[Figure]

P4 L30 small errors?

P11 L8-10 and P12 L4-5: I think these two sentences are inconsistent. Does ship-base observation underestimate or overestimate the variability of omega ?

---

## Referee Comment (RC3) · Anonymous Referee #3 · 10 May 2016

Sutton and co-authors introduce a valuable, impressive dataset and provide a useful analysis of the range of seasonal variability across moored timeseries for pH and Omega aragonite. They compare across 12 open ocean, coastal and coral reef locations. In section 2, they carefully assess sources of uncertainty and clearly describe the reasonable choices made in the omega calculations. The figures are clear. It's a very nice contribution, and I don't see a need for much revision prior to publication.

I would suggest the authors add some additional discussion of the degree to which we understand how organisms respond to variability outside the preindustrial range, as

this is a major focus in the analysis. They discuss a few oyster species with Figure 9 in section 3.2. I would suggest referring to this in the introduction, so as to better motivate the analysis. And also if there are other examples that could be used as motivation, that would be helpful. The motivation is presently termed primarily in terms of the general processes occurring in the coastal zone and the performance of numerical models.

A few additional, minor points:

Introduction, page 1, Line 32: Comment: 30% is not incorrect if one takes cumulative FF emission and cumulative land use emissions = cumulative anthropogenic emissions. However, with respect to land processes, this does ignore the fact that much of the cumulative terrestrial sink is, in fact, regrowth after previous land use clearing. And uncertainty is very large on the mean land use source. That the ocean has absorbed 41,48% of fossil fuel emissions (Ciais et al. 2013, Sabine et al. 2004) has less uncertainty. I suggest (but do not insist) the authors consider using either the 41 or 48% "of fossil fuel emissions" as this is better quantified.

Introduction, page 2, line 34: "is adapated" suggested phrasing

Figures 2-7: It would help to add some additional labeling for "Open Ocean", "Coastal", etc. This would help the reader to follow the discussion, as these figures do all look so similar otherwise. This information is provided already in the captions, so I am thinking of something bolder to stand out on the figure itself

---

## Author Response (AR1)

*Note to editors: Response to reviewers pages 1-4, then edited manuscript restarts at page 1 after that*

**Response to Anonymous Referee #1**

*We would like to thank anonymous referee #1 for her/his thoughtful review. Our responses to all of the referee's comments are italicized below.*

The study is an attempt to constrain the variability of surface ocean carbonate chemistry via compiling 3-hourly moored observations for 12 open ocean, coastal, and coral reef locations. Further, these present-day conditions are compared to biologically relevant thresholds associated with ocean acidification.

These are very relevant topics in the context of anthropogenic climate change and definitely within the scope of BG. The paper is detailed and well-written. I would appreciate a more thorough evaluation of state-of-the-art ESMs against this new set of observations, which would be very valuable (as the authors correctly note, ESMs still have issues in capturing the full magnitude of variability), but that is perhaps beyond the scope of this paper.

*In the observations-modeling comparisons within this study, our focus was to directly compare seasonal to interannual variability of surface ocean $\Omega_{aragonite}$ and pH. This limited our comparison to studies that presented modeling results of these parameters with statistics of annual amplitude and interannual variability (in this case, we presented SD of annual anomalies). The more recent ESM papers the referee mentions focus primarily on trends and lack these types of statistics on seasonal and interannual variability of $\Omega_{aragonite}$ and pH. While we did not focus on detection of long-term trends in this paper (because the paired pH observations are not long enough yet to interrogate trends), we do agree that we can make a general evaluation of pre-industrial vs. present day moored observations in the context of these more recent ESM studies that explore time of emergence of OA trends (e.g., Mora et al. 2013, Keller et al. 2014, and Rodgers et al. 2015). We also see that Rodgers et al. (2015) do present SD of linear trends of surface ocean $\Omega_{aragonite}$ (Fig A1[b]), which they attribute to the background natural variability (i.e., all temporal variability from sub-seasonal to decadal), so we plan to add a statement about how our open ocean observations compare to the variability presented in that study.*

p 1 line 18: Looking at "long-term change" in the context of "natural variability", wouldn't be the challenge the detectability (and correct estimation) of these long-term trends in OA - which then in turn affect marine life?

*Yes, we agree and will clarify that the challenge is in detecting and interpreting long term change in the context of natural variability.*

p 1 line 29: "Modes" refers to ENSO, NAO etc. - I would recommend the usage of "patterns" or something like that.

*We have changed modes to patterns throughout.*

p 1 line 35: Although - While?

*We have made this change.*

p 2 line 10: than in open..

*We have made this change.*

p 2 line 19ff:  Please also include more recent studies, e.g., Keller at al., 2014 and Rodgers et al., 2015.
*We agree to add a review of and reference to more recent modeling studies relevant to the scope of this study (see above response).*

p 3 line 15: "and to ground truth carbonate chemistry variability in earth system models." Quite strong wording for what is actually done later, also considering the small number of locations and (partly quite old) models. Rewrite.
*Thank you for pointing this out.  We agree to replace this by simply stating this study will compare mooring observations with some past modeling estimates of seasonal to interannual variability of Ωaragonite.*

p 5 line 1: "or variability" - delete. SD was introduced as variability just one sentence before.
*We have made this change.*

p 5 line 34: Q3 + 1.5 x IQR
*Very important catch! We have made this change.*

p 7 line +/-25:  How exactly do you define overlap?  At KEO/Ωarag, the whisker in November seems also to be in the gray.
*We identify overlap whenever present day monthly observations within 1.5xIQR (i.e., within whiskers of the box and whisker plots) overlap with pre-industrial range (defined on page 5 lines 21-37).  KEO Ωaragonite observations do overlap very slightly with pre-industrial bounds in November and December, and we will add this to the discussion you mention on page 7.*

p 8 line 9ff: This paragraph is not really clear, please rephrase.
*We have made this change.*

p 8 line 11: Chá b˜a is shown in Fig. 5, not 4.
*We agree this section was confusing and have simplified by focusing on the high subseasonal variability at Chaba.*

p 9 line 2: see p 1 line 29
*We have changed modes to patterns throughout.*

p 9 line 6: see p 1 line 29
*We have changed modes to patterns throughout.*

p 12 line 3: see p 1 line 29
*We have changed modes to patterns throughout.*

Tables & Figures:
The gray and blue features (cells, shades, grids) are hardly visible (printed out, the light gray in Tab. 4 is completely absent, as is the grid in Fig. 8). Please replace with stronger colors.
*We would appreciate the advice from Biogeosciences editors on this issue.  We attempted to maximize color differentiation in pdf format but, like the reviewer, have found less desirable results on different printers and projectors, so we're not sure what is best.*

**Response to Anonymous Referee #2**

*We would like to thank anonymous referee #2 for her/his thoughtful review. Our responses to all of the referee's comments are italicized below.*

The authors observed pH and pCO2 of surface seawater for several years by using moor system and calculated present-day monthly omega. Then, they discuss biological thresholds of shellfish using pre-industrial calculated omega. Overall, I agree with the authors. I also would like to know time series changes of pH and omega during observation, but this would be another topic. Since the manuscript is well written, only a few minor comments are attached below.

*We are also very interested in trends of pH and $\Omega$aragonite during the time series; however, because the pH time series are not long enough yet to interrogate these trends, we did not focus on detection of long-term trends in this paper.*

Fig.2-7 I would like you to add pCO2 data

*Because a major focus of this paper was to better understand present day exposure to known biological thresholds for shellfish larvae around coastal moorings, we focused on the parameters for which these thresholds have been established. We agree that pCO2 is also of interest to the ocean acidification community, but in order to maintain this focus and keep the paper to a reasonable length, we intend to present the pCO2 analysis is a separate paper.*

P4 L1 "Overall uncertainty..." Is it ture ? I think 2 uatm is too small.

*Yes, we have published the uncertainty assessment, which uses lab testing and comparisons to ship-based measurements in the field, in the following publication: Sutton et al. A high-frequency atmospheric and seawater pCO2 data set from 14 open-ocean sites using a moored autonomous system. Earth Syst. Sci. Data, 6, 353–366, 2014.*

P4 L30 small errors?

*These small errors could reflect the sensitivity of the pCO2-pH pairing to systematic errors and/or slight mismatches in time and space of the pCO2 and pH measurements (Cullison-Gray et al., 2011, Marine Chemistry 125: 82–90).*

P11 L8-10 and P12 L4-5: I think these two sentences are inconsistent. Does ship-base observation underestimate or overestimate the variability of omega ?

*Thank you for pointing out the inconsistency between these statements. We will add the clarification in the statement on page 12 lines 4-5 that ESMs and ship-based observations generally underestimate temporal variability of \*open ocean\* $\Omega$aragonite.*

**Response to Anonymous Referee #3**

*We would like to thank anonymous referee #3 for her/his thoughtful review. Our responses to all of the referee's comments are italicized below.*

Sutton and co-authors introduce a valuable, impressive dataset and provide a useful analysis of the range of seasonal variability across moored timeseries for pH and Omega aragonite. They compare across 12 open ocean, coastal and coral reef locations. In

section 2, they carefully assess sources of uncertainty and clearly describe the reasonable choices made in the omega calculations. The figures are clear. It's a very nice contribution, and I don't see a need for much revision prior to publication.

I would suggest the authors add some additional discussion of the degree to which we understand how organisms respond to variability outside the preindustrial range, as this is a major focus in the analysis. They discuss a few oyster species with Figure 9 in section 3.2. I would suggest referring to this in the introduction, so as to better motivate the analysis. And also if there are other examples that could be used as motivation, that would be helpful. The motivation is presently termed primarily in terms of the general processes occurring in the coastal zone and the performance of numerical models.

*This is a very useful perspective, and we agree additional background in the introduction on relevance to biological response would improve the paper. While our understanding of how organisms respond to variability outside pre-industrial conditions is limited, there are experimental studies that test organism response at pre-industrial, present day, and future ocean acidification conditions, which we can summarize to bolster the motivation.*

Introduction, page 1, Line 32: Comment: 30% is not incorrect if one takes cumulative FF emission and cumulative land use emissions = cumulative anthropogenic emissions. However, with respect to land processes, this does ignore the fact that much of the cumulative terrestrial sink is, in fact, regrowth after previous land use clearing. And uncertainty is very large on the mean land use source. That the ocean has absorbed 41,48% of fossil fuel emissions (Ciais et al. 2013, Sabine et al. 2004) has less uncertainty. I suggest (but do not insist) the authors consider using either the 41 or 48% "of fossil fuel emissions" as this is better quantified.

*Our estimate takes into consideration a compilation of inventories of anthropogenic carbon (observational and model-based estimates), which gives an estimate of the global ocean inventory in 2010 of 155±31 PgC (Khatiwala et al., 2013, www.biogeosciences.net/10/2169/2013/). When combined with the latest, annually-updated Global Carbon Project budget estimates for total cumulative emissions from 1870 to 2014 of 400±20 GtC from fossil fuels and cement and 145±50 GtC from land use change (Le Quéré et al., 2015), the total global ocean sink is ~30%. We believe this is the most comprehensive analysis to date that takes into consideration the entire global carbon budget. Now looking at this with a more critical eye, we should modify the citations for that statement to Le Quéré et al., 2015 and Khatiwala et al., 2013.*

Introduction, page 2, line 34: "is adapated" suggested phrasing
*We have made this change.*

Figures 2-7: It would help to add some additional labeling for "Open Ocean", "Coastal", etc. This would help the reader to follow the discussion, as these figures do all look so similar otherwise. This information is provided already in the captions, so I am thinking of something bolder to stand out on the figure itself
*We have made this change.*

[revised manuscript text omitted]